# Flow-Diverting Stents During Mechanical Thrombectomy for Carotid Artery Dissection-Related Stroke: Analysis from a Multicentre Cohort

**DOI:** 10.3390/brainsci15060629

**Published:** 2025-06-11

**Authors:** Osama Elshafei, Jonathan Cortese, Nidhal Ben Achour, Eimad Shotar, Jildaz Caroff, Léon Ikka, Cristian Mihalea, Vanessa Chalumeau, Maria Fernanda Rodriguez Erazu, Mariana Sarov, Nicolas Legris, Jean-Christophe Gentric, Frederic Clarençon, Laurent Spelle

**Affiliations:** 1NEURI Brain Vascular Center, Interventional Neuroradiology, Bicetre University-Hospital, 78 rue du Général Leclerc, 94270 Le Kremlin-Bicetre, France; osama_elshafei@mans.edu.eg (O.E.);; 2Department of Neurology, Faculty of Medicine, Mansoura University, Mansoura 35516, Egypt; 3Faculty of Medicine, Paris-Saclay University, INSERM U1195, 94276 Le Kremlin-Bicetre, France; 4Department of Radiology, University Hospital of Brest, 29609 Brest, France; 5Department of Neuroradiology, Pitié-Salpêtrière University Hospital, Sorbonne University, 75013 Paris, France; 6Department of Neurology, Bicetre University-Hospital, 94270 Le Kremlin-Bicetre, France

**Keywords:** flow-diverting stent, stroke, carotid artery dissection, mechanical thrombectomy

## Abstract

Background and Purpose: Mechanical thrombectomy in the context of internal carotid artery dissection (ICA-D) lesions is an undesirable procedure that may necessitate carotid stenting. Flow-diverting stents (FDSs) are promising devices with numerous advantages, particularly in cases involving tortuous anatomy. Here, we investigate the use of FDSs in the acute management of carotid dissection during mechanical thrombectomy procedures in patients with dissection-related strokes. Materials and Methods: This was a multicentric retrospective observational study of consecutive patients admitted for mechanical thrombectomy due to acute ischaemic stroke with ICA-D and treated with an FDS in the acute setting between July 2018 and February 2023. Patient records, procedural details, and post-procedural outcomes, including follow-up data, were reviewed. Results: A total of 11 patients (10 patients with unilateral ICA-D and one patient with bilateral ICA-D) were included, 10 of whom were male, with a median age of 54 years (range: 35–85 years) and NIHSS scores at admission ranging from 3 to 32 (median 13). Eight cases (73%) involved intracranial occlusion (tandem stroke), with the intracranial occlusion managed first each time. An FDS was selected when the dissection was long and/or the ICA was tortuous, and successful deployment was achieved in all patients with a favourable angiographic outcome (TICI 2B-3). A favourable outcome (modified Rankin scale 0–2 at 90 days) was observed in five patients (45%), with four patients (36%) experiencing symptomatic ICH and three patients having stent occlusion out of the 12 treated ICA-D cases. Conclusions: The use of FDSs for acute stenting in ICA-D-related stroke can be performed efficiently, resulting in excellent angiographic outcomes and an acceptable rate of favourable outcomes specific to the pathology. Larger prospective studies are still needed to confirm the potential benefits of FDSs in acute situations.

## 1. Introduction

Internal carotid artery dissection (ICA-D) accounts for approximately 2–2.5% of acute ischaemic strokes (AISs), with this percentage rising to around 10–25% in patients younger than 45 years old [1]. Dissection occurs due to idiopathic (spontaneous), traumatic, or iatrogenic causes, leading to the formation of an intimal tear and arterial wall separation by the formation of an intramural haematoma secondary to vasa vasorum rupture [2,3].

Approximately half of all ICA-D cases result in an AIS, with this being the initial presentation in about one-third of ICA-D patients [4]. Strokes occurring in the context of steno-occlusive ICA-D can be attributed to either haemodynamic insufficiency (especially if there is inadequate collateral flow) or, more commonly, artery-to-artery thromboembolism [5]. The latter is referred to as a tandem occlusion; ICA-D is responsible for 20% to 30% of tandem strokes, which have the worst prognosis among all stroke types [6,7].

The management of tandem lesions with intravenous thrombolysis alone is associated with a poor functional outcome in up to 80% of patients [8]. Mechanical thrombectomy (MT) is the standard of care for tandem occlusions; however, the optimal strategy to manage carotid dissection remains controversial [9,10]. Treatment strategies include retrograde, antegrade, and delayed approaches. Another point of contention is whether to use a carotid stent or balloon angioplasty for the extracranial occlusion, as well as the anticoagulation and antiplatelet regimens for the treatment. It is also necessary to take into consideration the choice of strategy, which is often left to the discretion of the treating operator; the need to adapt to the specific scenario; the underlying cause (atherosclerosis versus dissection); the stroke severity; the infarct size; and the vascular anatomy. Therefore, the approach to MT in tandem occlusions should be tailored on an individual operator- and patient-specific basis [11].

ICA-D can be managed by carotid artery stent placement using devices such as the Carotid Wallstent (Boston Scientific, Marlborough, MA, USA), Precise (Cordis, Fremont, CA, USA), or Cristallo Ideale (Medtronic, Kalamazoo, MI, USA), which are relatively stiff and easier to place in straight segments [12]. The use of flow-diverting stents (FDS) offers an interesting alternative in situations where the dissection includes a tortuous segment of the ICA due to their flexibility and the size of the microcatheters used to deploy them.

Despite the advantages of FDSs, the literature on their use during MT for ICA-D-related stroke is scarce. To date, four studies have been published: two case reports and two case series, comprising a total of 21 patients (Table 1). Herein, we investigate the efficacy and safety of using FDSs in ICA-D-related stroke.

## 2. Materials and Methods

The Institutional Review Board (CERIM) approved this study (code number CRM2402-394) in February 2024. All study protocols and procedures were conducted in accordance with the Declaration of Helsinki. The need for written informed consent for this study was waived because it was a non-interventional retrospective study of routinely acquired data. Patients were informed that they could oppose the use of their health-related data for research purposes.

This was a multicentre retrospective observational study; 3 primary stroke centres in France participated (NEURI Brain Vascular Center at Bicetre University Hospital, Department of Radiology at University Hospital of Brest, and Department of Neuroradiology, Pitié-Salpêtrière University Hospital), and all consecutive patients were included from July 2018 to February 2023 if they met the following inclusion criteria: AIS due to ICA-D (with or without concomitant intracranial large-vessel occlusion) referred for a mechanical thrombectomy, ICA-D confirmed with digital subtraction angiography evaluation, and at least one FDS used to treat the dissection in the acute setting. Stenting was only performed after confirmation of the absence of intracranial bleeding with CBCT due to the need for antiplatelet treatment.

Baseline and follow-up data: Demographic data including age, sex, and vascular risk factors were collected. Clinical data including the National Institutes of Health Stroke Scale (NIHSS) score were assessed on admission and at 24 h after the procedure, as well as the modified Rankin score (mRS) after 90 days.

Radiological and angiographic assessment data: The imaging type, the Alberta Stroke Program Early CT Score (ASPECTS) initially and 1 day after the procedure, and the presence of an LVO and its location. In DSA, we assessed the occlusion site, the initial and final mTICI scores, and the FDS patency.

Endovascular procedures: The endovascular treatment technique was chosen by the neuro-interventional radiologist performing the procedure. Procedural data including the type of anaesthesia, onset-to-puncture time and procedural duration, adopted strategy (proximal to distal or distal to proximal), and use of antiplatelet and thrombectomy techniques were collected.

Adverse events: Events including symptomatic intracranial haemorrhage (defined by the presence of a haemorrhagic infarction or a parenchymal haemorrhage confirmed by CT or MRI associated with NIHSS deterioration), embolisation to distal or new territories, or procedural complications were recorded.

Statistical analysis: Descriptive data are reported as the median (range) or mean ± SD for continuous variables and as frequencies (%) for categorical variables.

## 3. Results

### 3.1. Demographics and Clinical Presentation

A total of 11 patients were studied (10 male patients [91%]); the median age was 54 years (range: 35–85 years). Six patients (55%) had at least one cardiovascular risk factor, one (9%) had atrial fibrillation, and only one patient had a history of taking antiplatelets. The baseline mRS was 0 in all patients except one (mRS 1).

Ten patients (91%) were admitted in the context of AIS suspicion directly to the primary stroke centre. One patient was already hospitalised at a tertiary hospital due to suspicion of a left carotid dissection without an AIS confirmed by initial MRI; the patient was placed under dual antiplatelet treatment with aspirin and ticagrelor, but he presented within the first 12 h with a sudden right hemiplegia with aphasia revealing a left T-carotid occlusion, with an AIS confirmed by a second MRI scan. The patient was subjected to drip and ship for a thrombectomy. The median NIHSS at admission was 13 (range: 3–32). See Table 2.

### 3.2. Baseline Imaging

Most patients (nine patients [82%]) were evaluated with MRI/angio-MR, including two patients that needed a complementary computed tomography angiogram (CTA), while two patients (18%) had a CT/CTA evaluation alone. The occlusion/dissection was located on the left side for five patients (45%) and on the right side for five patients (45%), and it was bilateral for one patient (9%).

The median baseline ASPECT and ASPECT-DWI scores were 8 (range: 6 to 10). Eight patients (73%) had concomitant intracranial occlusions (tandem stroke).

### 3.3. Endovascular Procedure

Seven patients (64%) were placed under general anaesthesia from the start, and, in three patients (27%), the procedure started with local anaesthesia and then was converted to general anaesthesia, while one patient (9%) remained under conscious sedation. The median onset-to-puncture time was 413 min (6 h and 53 min), and the median procedural duration was 145 min (2 h and 25 min).

The angiographic assessment confirmed the 12 ICA-Ds: two (17%) limited to the cervical segment, six (50%) extending to the petrous segment, three (25%) to the cavernous segment, and one (8%) to the intracranial segment. High carotid tortuosity (e.g., carotid loop) was also confirmed in 6/12 ICA-D cases (50%).

The adopted strategy for all tandem occlusions was to manage the intracranial occlusion first and then the carotid (distal to proximal). For the intracranial thrombectomy, a combined technique was used in six patients and aspiration alone in one patient, while, in one patient, an intracranial thrombectomy was not needed due to recanalisation. Successful recanalisation (final TICI 2b, 2c, or 3) was achieved in all patients (100%).

FDSs were successfully deployed in the carotid artery in all patients. In five ICA-Ds (45%), the FDS was used in combination with a carotid Wallstent proximally. The median number of stents implanted was two (range: 1–4) per ICA-D. In all patients, the carotid was effectively reconstructed with 100% patency. Figure A1 presents a case illustration of the procedure. See Table 3.

### 3.4. Thrombolysis and Antiplatelet Management

Seven patients (64%) received intravenous thrombolysis before being transferred to mechanical thrombectomy. Besides the one patient that was placed under aspirin and ticagrelor previously, all patients were prepared for stenting using an IV bolus injection of aspirin (250 mg to 500 mg). Ticagrelor was used as a second antiplatelet treatment in five patients (45.5%), cangrelor in five patients (45.5%), and integrilin in one patient (9%). The second antiplatelet treatment was started in most cases just before stent placement and after flat-panel CT confirming the absence of haemorrhagic transformation, while, in two cases with ticagrelor, it was started after a control CT confirming no haemorrhagic transformation. In every case with cangrelor, relay was performed at day 1 with ticagrelor. Upon discharge, the dual antiplatelet therapy consisted of aspirin and ticagrelor for 6 months, followed by aspirin alone.

### 3.5. Complications and Outcomes

Four patients (36%) experienced a symptomatic intracranial haemorrhage with NIHSS worsening (ECASS score: PH2 in two patients and HI2 in two patients). Two patients (16%) experienced stent occlusion: one occurred after 24 h, where the patient was on integrilin, which was stopped due to haemorrhagic complications; and one patient experienced occlusion at day 7 after antiplatelet therapy was stopped for a rescue craniectomy at day 3. All other stents remained patent.

The day-1 NIHSS ranged from 0 to 23 (median 12.5) and the day-1 ASPECT score ranged from 3 to 9; the median was 8. The 3-month mRS median was 2.5 and ranged from 2 to 6, with favourable outcomes (mRS 0–2) in 5/10 patients (50%; one with missing data).

## 4. Discussion

In this study, we investigated the use of FDSs in ICA-D-related stroke during the acute stage. We retrospectively included 12 patients. Our study showed a 100% success rate (11 out of 11 patients) for FDS placement in the ICA-D during acute stroke management, with a 100% successful recanalisation rate.

The efficacy and safety of FDSs in the treatment of intracranial aneurysms have been demonstrated [16,17,18]. They offer the advantage of reconstructing the pathological arterial segment, providing a scaffold for neointima formation while redirecting the blood flow to the parent artery [19]. New off-label indications have emerged for the use of FDSs, especially in extracranial carotid dissections, with or without a pseudoaneurysm, with satisfactory outcomes [20,21,22,23]. Operators’ decision to use FDSs in ICA-D-related stroke is based on some unique features of FDSs. An FDS allows the navigation of a microcatheter through the dissection, even in the presence of a highly tortuous anatomy, including a cervical carotid loop and the petrous/cavernous segments of the ICA. Additionally, FDSs are self-expanding, with less radial force than conventional stents, thus reducing the risk of thromboembolism in the initial deployment. Furthermore, the high metal coverage surface of FDSs allows for tighter, secure clot containment outside the stent, and their low porosity prevents in-stent clot extension [20]. Finally, FDSs facilitate easier navigation of balloon catheters for remodelling or angioplasty through their lumen [10].

Generally, acute stenting carries high rates of technical success, but it is associated with variable rates of haemorrhage, ranging from 18% to 43%, and in-stent thrombosis in up to 17% [24,25,26]. These potential complications have led to practice variations and clinical uncertainty about the optimal approach, with no studies demonstrating the superiority of acute stenting versus delayed stenting [27]. Interestingly, in this study using FDSs, which contain more metal than the typical carotid stents, we found similar rates of haemorrhage and stent occlusion.

The major concern in acute carotid stenting during MT is haemorrhagic complications, especially if patients have already received intravenous thrombolysis. Additionally, the use of antithrombotic drugs during stenting may increase this risk. Nonetheless, antithrombotic drugs are important, as most thrombotic events occur within hours of the acute stenting procedure, and they are associated with a higher rate of intracranial recanalisation (83%) and a higher rate of favourable clinical outcomes (58%) compared to not using antithrombotic drugs [13]. The optimal antithrombotic management of stenting in acute stroke patients is still unclear, and the choice of timing and the nature of the intravenous antithrombotic agents used is left at the discretion of the physician [10].

The use of FDSs for the acute management of ICA-D-related stroke has been scarcely reported thus far, as shown by our systematic review of the literature. Gramegna et al. reported the successful use of an FDS in one patient with an acute ischaemic stroke due to cavernous to supraclinoid segment dissection [14]. Similarly to our results, another case series reported a high rate of technical success, with 66% good clinical outcomes [10]. Anther case series reported the reconstruction of an ICA (loop) dissection with an FDS in four cases during mechanical thrombectomy. A combination of FDSs and carotid stents was used in three cases, and an FDS alone was used in the fourth case, all resulting in good angiographic and clinical outcomes [15]. In another case report on cervical ICA occlusive dissection reconstruction, a PED FDS followed by reocclusion and a carotid Wallstent and two Precise Pro nitinol stents were deployed in a telescoping fashion from the common carotid artery into the PED [28].

Regarding the limitations of this study, the results are challenged by its retrospective nature, which may have introduced selection bias. Additionally, the relatively small number of patients included is primarily attributed to the limited availability of patients and the recent adoption of the approach proposed here. The choice of a specific stenting approach was left to the discretion of the operator, influenced by their expertise and personal preferences; furthermore, antiplatelet therapy was not standardised, which limits the application of these results to larger populations. Therefore, prospective trials are needed to validate these findings.

## 5. Conclusions

This small case series demonstrated the safety and technical feasibility of using FDSs for acute stenting in ICA-D-related stroke, particularly in cases of tortuous anatomy. It resulted in excellent angiographic outcomes and a relatively acceptable rate of favourable outcomes specific to the pathology. However, larger prospective studies are essential to validate the potential benefits of FDSs in acute settings.

## Figures and Tables

**Table 1 brainsci-15-00629-t001:** Published studies on the use of flow diverters for internal carotid artery dissection.

Author	Year	Number of Patients	Final mTICI 2b-3	sICH	mRS ≤ 2 at 3 Months	APT Protocol	Stent Patency at 3 Months
Gramegna LL, Cardozo A, Folleco E, et al. [13]	2021	1	NA	No	100% (1/1)	Aspirin (900 mg, intravenous) and tirofiban (48 mL/h, intravenous). At 24 h post-procedure, aspirin (100 mg) and clopidogrel (75 mg) were started for 3 months.	Achieved 60–70% right bulb ICA stenosis and greater than 70% stent stenosis of the FDS. The right bulb ICA stenosis was successfully treated with a carotid stent, and the in-stent stenosis was treated with balloon angioplasty, achieving good expansion with residual in-stent stenosis of less than 30%.
Diaz-Silva H, Piñana C, Gramegna LL, et al. [10]	2021	12	100%	8% (1/12)	66% (8/12)	Nine patients were loaded with inyesprin and tirofiban, 2 patients received monotherapy with tirofiban or inyesprin alone, and 1 patient received inyesprin, abciximab, and heparin.	100% (17% [2/12], one > 50% stenosis and one < 50% stenosis).
Kühn AL, Singh J, Massari F, et al. [14]	2022	7	100%	No	80% (4/5)	Received 600 mg aspirin per rectum immediately after the procedure and were started on aspirin 81 mg or 325 mg and Plavix 75 mg after 24 h. One patient received an eptifibatide bolus intraoperatively.	100%
Befera N, Griffin AS, Hauck EF [15]	2018	1	100%	No	100% (1/1)	Dual-antiplatelet therapy (325 mg rectal aspirin and 300 mg oral clopidogrel) and full heparinisation were initiated before the procedure. Then, dual-antiplatelet therapy was continued with 81 mg aspirin and 75 mg clopidogrel for three months.	100%

NA: not available.

**Table 2 brainsci-15-00629-t002:** Patients’ baseline characteristics.

Patient	Sex	Age	Side	Initial NIHSS	Initial ASPECTS	Intracranial Occlusion Location	Dissection Location	rt-PA
1	M	44	Right	9	7	M2	Cervical to petrous	Yes
2	M	85	Right	12	9	T-Carotid	Cervical to intracranial	Yes
3	M	73	Right	15	6	M1	Cervical to petrous	No
4	M	42	Left	17	7	M2 + A2	Cervical to cavernous	Yes
5	M	73	Left	12	10	T-Carotid	Infrapetrous to petrous	Yes
6	M	35	Right	13	10	None	Cervical	Yes
7	M	43	left	3	10	None	Cervical to petrous	No
8	F	51	Left	5	9	M2	Cervical	Yes
9	M	56	Bilateral	32	8	None	Cervical to petrous (both sides)	No
10	M	54	Left	14	6	M1	Cervical to Cavernous	Yes
11	M	56	Right	14	7	M1	Infrapetrous to cavernous	No

**Table 3 brainsci-15-00629-t003:** Procedure characteristics and outcomes.

Patient	Onset to Puncture (min)	Procedure Duration	MT Technique	Final TICI	FDS Used (*n*)	Second Stent	sICH	24 h ASPECTS	Discharged NIHSS	mRS	Last FU (Month)	APT Bolus	Postop APT	Stent Patency
1	712	120	None	2B	Vantage (1)	No	No	7	13	NA	NA	Cangrelor	Ticagrelor + Aspirin	NA
2	614	210	Combined	2C	Vantage (1)	Wallstent	Yes	7	20	6	NA	Cangrelor	Ticagrelor + Aspirin	NA
3	300	480	Combined	2B	Vantage (1)	Wallstent	No	3	15	5	0.3	Integrilin	Ticagrelor + Aspirin	Occlusion day 1
4	540	140	Combined	3	Vantage (3)	No	Yes	8	23	2	12	DAPT	Ticagrelor + Aspirin	Patent
5	390	60	CA	3	Silk (1)	No	No	6	12	3	3	Aspirin + Tirofiban	Ticagrelor + Aspirin	Patent
6	255	150	None	2C	Vantage (1)	No	No	8	12	2	3	Cangrelor	Ticagrelor + Aspirin	Patent
7	NA	NA	None	2C	PED (2)	Wallstent	No	8	1	3	10	Reopro	Ticagrelor + Aspirin	Patent
8	117	157	Combined	2B	PED (1)	Wallstent	No	9	2	2	60	Aspirin + Cangrelor	Ticagrelor + Aspirin	Patent
9	1950	240	None	3	PED (2 left + 2 right)	No	No	8	0	2	5	Aspirin + Cangrelor	Ticagrelor + Aspirin	Patent
10	300	85	Combined	2C	PED (2)	Wallstent	Yes	6	16	4	3	Cangrelor	Ticagrelor + Aspirin	Occlusion day 7
11	436	63	Combined	2B	PED (1) + Surpass Evolve (1)	No	Yes	8	NA	2	6	Aspirin	Ticagrelor + Aspirin	Patent

DAPT: dual antiplatelet therapy.

## Data Availability

The study data are available from the corresponding author upon reasonable request due to institutional regulations on patient data.

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
