# Peer review of "Flow-Diverting Stents During Mechanical Thrombectomy for Carotid Artery Dissection-Related Stroke: Analysis from a Multicentre Cohort"

_brainsci, 2025, doi:10.3390/brainsci15060629_

Round 1
Reviewer 1 Report
Comments and Suggestions for Authors
Thank you for the opportunity to review the manuscript. I would like to congratulate the authors for their substantial efforts in conducting this study.
Endovascular treatment of stroke caused by carotid artery dissection is a very interesting and worthwhile topic to analyze. The author(s) assessed the endovascular management of internal carotid artery dissections with flow diverters (FDs). This topic, though not entirely original, remains highly relevant to the field, as dissecting tandem occlusions are relatively rare, and clear treatment recommendations are still lacking.
The authors describe 11 cases treated with FDs, which constitutes one of the larger series published to date. Moreover, they provide detailed information on the antiplatelet therapy administered to patients, enhancing the clinical significance of the manuscript.
The introduction effectively outlines the rationale for the study and smoothly guides the reader into the core of the manuscript. The Materials and Methods section is well-written, and the Results section is further enriched by descriptive tables.
The Discussion section is engaging, though in my opinion, it could benefit from a more detailed explanation or comparison: Why would FDs be superior to classic intracranial stents? The latter are 2–3 times cheaper, available in a wider variety of sizes, and possess similar radial force and comparable treatment effects—for example, Leo+ stents, which appear to be widely used in such cases, come in lengths of up to 75 mm and diameters of up to 5.5 mm. Several case series involving classic intracranial stents have already been published.
The conclusions are consistent with the evidence presented and address the main research question. The authors acknowledge the study's limitations—its retrospective design and the limited number of cases—and suggest the need for a prospective trial. However, given the rarity of such cases, the feasibility of such a trial is likely low.
Nevertheless, I commend the authors for their excellent idea and well-executed study. I recommend this manuscript for publication.
The references are appropriate and well-selected.
Author Response
Comments 1: Why would FDs be superior to classic intracranial stents? The latter are 2–3 times cheaper, available in a wider variety of sizes, and possess similar radial force and comparable treatment effects—for example, Leo+ stents, which appear to be widely used in such cases, come in lengths of up to 75 mm and diameters of up to 5.5 mm. Several case series involving classic intracranial stents have already been published. |
Response 1: Thank you for your valuable comment regarding the role of classic stents such as Leo+ in certain clinical scenarios. We fully acknowledge the importance of classic stents for distal lesions, cost-sensitive cases, and situations where operator familiarity is a significant consideration. However, the primary scope of our paper is to present the role and and outcomes specifically with flow diverter stents (FDS). |
Reviewer 2 Report
Comments and Suggestions for Authors
This is an interesting case series and literature review that addressed a very relevant topic of use flow diverting stent in treatment of acute dissection within internal carotid artery resulting in ischema. Though the series is not large many of the relevant variables that are of concern to successful recanalization and maintenance of arterial patency while avoiding and adressing hemorrhagic conversion has been appropriately abstracted and followed. This series shows that a very high success rate with appropriate clinical efficacy and acceptable risk profile of such treatment strategy in addressing acute ischemic stroke from large vessel occlusion or stenosis.
There are a issues that should be addressed:
- Was this strategy only utilized in absence of any blood detected on cone beam CT or was that per discretion of the interventionalist?
- Was there a uniform approach or any specific criteria for utilizing balloon remodeling following flow diversion
- Was addition other non flow diverting stent based on a criteria or uniform approach for this series
- Are the specific flow diverting stent utilized in this study known and could the authors present that and potentially comment on the radial force of each used and how that has played into the success of reconstruction following dissection
- Was there a uniform approach for following patency of the affected artery post intervention in this series
- Was there P2y12 activity checked in all patients after and more specifically in patients with occlusion or hemorrhagic conversion
- What was the strategy of antiplatelet use after suspected hemorrhagic conversion and was this a uniform approach.
- on page 7 line 189 authors report that figure 2 is representative of their PRISMA flow chart but they are referring to figure 1
- In the results presented in table 3 from their systematic review of the literature, there is a very high rate of longterm residual patency of the arteries despite seemingly in adequate antithrombotic use which is unlike the data presented from the authors own series. That fact should be mentioned and discussed in the discussion section.
Author Response
Comments 1: Was this strategy only utilized in absence of any blood detected on cone beam CT or was that per discretion of the interventionalist? |
Response 1: Thank you for pointing this out. Due to the retrospective nature of the study, the case selection was done by the interventionalist. However, as a “rule of thumb” stenting is only performed after confirmation of the absence intracranial bleeding with CBCT due to the need of antiplatelet treatment. We have modified our methods
|
Comments 2: Was there a uniform approach or any specific criteria for utilizing balloon remodeling following flow diversion |
Response 2: Thank you for your comment. Due to the retrospective nature of the study, angioplasty was performed at the discretion of the operator. It was performed in case of incomplete opening of the stents.
Comments 3: Was addition other non-flow diverting stent based on a criteria or uniform approach for this series Response 3: Thank you for your comment. Yes, flow diverters are used in loop or tortuous anatomy but if longer stent material was needed in a straight segment, operators often switch to a cheaper device.
Comments 4: Are the specific flow diverting stent utilized in this study known and could the authors present that and potentially comment on the radial force of each used and how that has played into the success of reconstruction following dissection Response 4: Thank you for your comment. Yes the utilized FDS for each patient were mentioned in table 2.
Comments 5: Was there a uniform approach for following patency of the affected artery post intervention in this series Response 5: Thank you for your comment. No, no uniform approach was used. Detailed follow up is described. Comments 6: Was there P2y12 activity checked in all patients after and more specifically in patients with occlusion or hemorrhagic conversion Response 6: Thank you for your comment. No, P2y12 activity was not checked but Ticagrelor was more used so it was not needed.
Comments 7: What was the strategy of antiplatelet use after suspected hemorrhagic conversion and was this a uniform approach Response 7: Thank you for your comment. In all cases of bleeding, antiplatelet treatment was interrupted and management was used on a case-by-case strategy.
Comments 8: on page 7 line 189 authors report that figure 2 is representative of their PRISMA flow chart but they are referring to figure 1 Response 8: Thank you for pointing this out. We accordingly corrected this mistake
Comments 9: In the results presented in table 3 from their systematic review of the literature, there is a very high rate of longterm residual patency of the arteries despite seemingly in adequate antithrombotic use which is unlike the data presented from the authors own series. That fact should be mentioned and discussed in the discussion section. Response 9: Thank you for pointing this out. Literature review is very limited and bias of publication may explain this discrepancy. We discussed this point now.
|
Reviewer 3 Report
Comments and Suggestions for Authors
First of all, thank you for the opportunity to review this manuscript.
Without a doubt, this is a very relevant topic in connection with the increase in mortality and disability, especially at a young age, due to thrombosis.
The introduction is very short.
Management of tandem lesions with intravenous thrombolysis is associated with a poor functional outcome in up to 80% of patients. [Rubiera M., Ribo M., Delgado-Mederos R., et al., Tandem internal carotid artery/middle cerebral artery occlusion: an independent predictor of poor outcome after systemic thrombolysis. Stroke, 2006. 37(9): p. 2301-2305]. The reference to this information is out of date (2006).
“Mechanical thrombectomy (MT) is the standard of care for tandem occlusions; however, the optimal strategy to manage carotid dissection remains controversial. “Please add more information to clarify the controversial.
“Therefore, the approach to MT in tandem occlusions should be tailored on an individual operator- and patient-specific basis”. This phrase is not clear.
The purpose of the study is stated briefly and specifically.
Please indicate the number and date of the approval document. Please indicate the name of the institute's ethics committee.
“This is a multicentre retrospective observational study; 3 primary stroke centers in France participated”. Please clarify which centers participated in this study.
Please explain in detail the radiographic and angiographic evaluation provided.
Please add inclusion and exclusion criteria.
The research materials must be listed point by point.
The results of the study should be made clearer by illustrations.
The discussion is devoted to general issues unrelated to the findings. In places it seems to be a continuation of the introduction.
It seems to me that the brief review of the literature violates the integrity and unity of the manuscript. In my opinion, it would be better if this section were combined with the introduction. In connection with this, it is necessary to change the title of the article.
Author Response
Comments 1: Management of tandem lesions with intravenous thrombolysis is associated with a poor functional outcome in up to 80% of patients. [Rubiera M., Ribo M., Delgado-Mederos R., et al., Tandem internal carotid artery/middle cerebral artery occlusion: an independent predictor of poor outcome after systemic thrombolysis. Stroke, 2006. 37(9): p. 2301-2305]. The reference to this information is out of date (2006). |
Response 1: Thank you for pointing this out. We had updated this reference
|
Comments 2: “Mechanical thrombectomy (MT) is the standard of care for tandem occlusions; however, the optimal strategy to manage carotid dissection remains controversial. “Please add more information to clarify the controversial. |
Response 2: Thank you for your comment. We accordingly added Treatment strategies including retrograde, antegrade and delayed approaches. Another controverse is whether carotid stent or balloon angioplasty for the extracranial occlusion. Also the anticoagulation and antiplatelet regimens for the treatment.
Comments 3: “Therefore, the approach to MT in tandem occlusions should be tailored on an individual operator- and patient-specific basis”. This phrase is not clear. Response 3: Thank you for your comment. We added And taking in consideration the choice of strategy which is often left to the discretion of the treating operator, the need to adapt to the specific scenario, the underlying cause (atherosclerosis versus dissection), stroke severity, infarct size and vascular anatomy
Comments 4: Please indicate the number and date of the approval document. Please indicate the name of the institute's ethics committee. Response 4: Thank you for your comment. We added the approval statement Institutional Review Board (CERIM ) approved this study (code number CRM2402-394) in February 2024.
Comments 5: “This is a multicentre retrospective observational study; 3 primary stroke centers in France participated”. Please clarify which centers participated in this study. Response 5: Thank you for your comment. We added the centers names to the manuscript (NEURI Brain Vascular Center in Bicetre University-Hospital, Department of Radiology in University Hospital of Brest and Department of Neuroradiology, Pitié-Salpêtrière University Hospital)
Comments 6: Please explain in detail the radiographic and angiographic evaluation provided. Response 6: Thank you for your comment. Radiographic and angiographic following centers protocol in accordance with international standard of care.
Comments 7: Please add inclusion and exclusion criteria Response 7: Thank you for pointing this out. This was a retrospective study and we included all consecutive patients were included from July 2018 to February 2023 if they had the following inclusion criteria: AIS due to ICA-D (with or without concomitant intracranial large vessel occlusion) referred for a mechanical thrombectomy, ICA-D was confirmed with digital subtraction angiography evaluation and at least one FDS was used to treat the dissection in the acute setting.
Comments 8: The research materials must be listed point by point. Response 8: Thank you for your comment. Methods have been clarified.
Comments 9: The results of the study should be made clearer by illustrations.. Response 9: Thank you for pointing this out. An illustrative figure is provided.
Comments 10: The discussion is devoted to general issues unrelated to the findings. In places it seems to be a continuation of the introduction Response 10: Thank you for pointing this out. We added some key point related to other comments.
Comments 11: It seems to me that the brief review of the literature violates the integrity and unity of the manuscript. In my opinion, it would be better if this section were combined with the introduction. In connection with this, it is necessary to change the title of the article. Response 11: Thank you for pointing this out. On the contrary we believe that literature review depicts how there is very few reports published. We hope this work will be a valuable addition before further studies |
Round 2
Reviewer 3 Report
Comments and Suggestions for Authors
First of all, thank you very much for the opportunity to read this manuscript. Indeed, many changes were made to the introduction, materials and methods. In my opinion, all these changes improved the manuscript and made it clearer and more understandable.
The authors answered all my questions clearly and convincingly; in fact, there are no comments on the research methodology or its results.
One outstanding issue remains: the systematic review presented in Table 3. In my opinion, combining a systemic review with a case report may reduce the identity of the manuscript.
Including this table in the introduction may strengthen the manuscript. Just don't mention that this is a systematic review. Simply state that the following sources were found in the literature that address the problem. It is important to change the title by removing "and Review of the Literature".
Author Response
Comments 1: the systematic review presented in Table 3. In my opinion, combining a systemic review with a case report may reduce the identity of the manuscript.
Including this table in the introduction may strengthen the manuscript. Just don't mention that this is a systematic review. Simply state that the following sources were found in the literature that address the problem. It is important to change the title by removing "and Review of the Literature".
Response 1: Thank you for your comment. We now decided to remove the systematic review from our title, methods and results and simply focus on introducing the existing literature with table in the introduction.